# The Association of Resilience with Way of Coping, Psychological Well-Being and Quality of Life in Parents of Children with Cancer

**DOI:** 10.3390/ijerph20105765

**Published:** 2023-05-09

**Authors:** Joyce Oi Kwan Chung, William Ho Cheung Li, Laurie Long Kwan Ho, Ankie Tan Cheung

**Affiliations:** 1School of Nursing, The Hong Kong Polytechnic University, Hong Kong, China; 2School of Nursing, The Nethersole School of Nursing, The Chinese University of Hong Kong, Hong Kong, China; williamli@cuhk.edu.hk (W.H.C.L.); longkwanho@cuhk.edu.hk (L.L.K.H.); ankiecheung@cuhk.edu.hk (A.T.C.)

**Keywords:** cancer, children, parents, psychological well-being, resilience, quality of life

## Abstract

Evidence shows that resilience is crucial to maintain psychological well-being and quality of life in the face of stress and adversity. However, the relationships between resilience and psychological well-being and factors associated with quality of life in Hong Kong Chinese parents of children with cancer are underexplored. This study aimed to examine the interrelationships among resilience, ways of coping, psychological well-being, and quality of life among Chinese parents of children with cancer, and identify factors associated with their quality of life. A cross-sectional study was conducted with 119 Chinese parents of children with cancer at the Hong Kong Children’s Hospital between January 2020 and March 2022. Parents’ resilience level, ways of coping, depressive symptoms, state anxiety scores, perceived social support, and quality of life were assessed. Participating parents (*n* = 119) included 98 mothers (82.4%) and 11 parents were from single-parent families (9.2%). Almost half (47.9%) of the parents were potentially at risk for depression. The results showed that participants from single-parent families reported statistically significantly lower levels of resilience (*p* < 0.001), more depressive symptoms (*p* < 0.001), and poorer quality of life (*p* < 0.001) than those who lived with their partners (married). In addition, parents who adopted problem-focused coping strategies reported statistically significantly higher levels of resilience (*p* < 0.001), fewer depressive symptoms (*p* < 0.001), and better quality of life (*p* < 0.001) than those who adopted emotion-focused coping strategies. A multiple regression analysis revealed that resilience (*p* < 0.001) was associated with quality of life among parents of children with cancer. This study provides further support that resilience is an important factor associated with quality of life in parents of children with cancer. Assessing resilience in parents is an important prerequisite for designing appropriate interventions to increase their resilience and enhance their quality of life.

## 1. Introduction

A diagnosis of cancer undermines children’s physical and psychological well-being and creates overwhelming psychological distress for their parents [1,2,3]. As caregivers, parents carry major burdens, both physically and psychologically, in taking care of their child throughout the cancer trajectory [4]. Furthermore, worry about the negative sequelae of cancer treatment and fear of losing a child may cause constant psychological distress for parents including anxiety and depression, which severely jeopardizes their quality of life [5]. A meta-analysis that determined psychological well-being in 9262 parents of children with cancer across 14 countries showed that parents had a high pooled prevalence of anxiety (21%) and depression (28%), which were significantly higher than among parents of children without cancer [6].

An increasing number of studies have examined families’ resilience in adaptation to cancer [7]. Resilience is defined as an individual’s ability to use personal and social resources to maintain psychological well-being in the face of stress and adversity [8,9]. A study that examined resilience in parents of children with cancer in the United States between 2009 and 2010 revealed that parents with low resilience resources were at high risk for poor psychosocial outcomes [10]. Another study found a positive relationship between resilience and quality of life in patients with a drainage enterostomy, with resilience being a crucial factor for enhancing psychological well-being and quality of life [11]. 

Although resilience has been associated with psychological outcomes and quality of life, the majority of previous studies were conducted in the United States or other Western countries [8,9,10,11,12]. It is therefore uncertain whether such relationships can be applied to different cultural groups such as parents of children in Hong Kong. Evidence suggests that culture has a significant impact on the relationship between resilience and quality of life [13]. In addition, Chinese people are influenced by the philosophical concepts of Confucianism and the associated notion of fatalism [14]. Some Chinese parents may therefore be inclined to believe that their child is “doomed” to have cancer, and that there are few ways to change that fate; such beliefs affect their ways of coping and resilience to adversity. A review of the literature including PubMed, Cochrane Library, or the Clinical Trials Registries (https://clinicaltrials.gov/ and https://www.isrctn.com/) revealed no previous studies examining resilience levels and influencing factors among Chinese parents of children with cancer in Hong Kong. This study aimed to examine the relationships among resilience, ways of coping, psychological well-being, and quality of life among Chinese parents of children with cancer. Quality of life is defined as an individual’s perception of their status in life, including physical and psychological status, degree of independence, and social relationships [15]. The National Cancer Institute emphasizes the importance of assessing and evaluating an individual’s quality of life during and after active treatment [16], which is crucial to better understanding the impact of cancer and its treatment on the child from a parent’s perspective. Improving the quality of life for parents of children with cancer should be the prime focus of healthcare professionals. Following the conceptual framework of previous literature [10,11,12,13], it was hypothesized in this study that resilience is associated with the quality of life among parents of children with cancer in this study (primary outcome).

## 2. Materials and Methods

### 2.1. Design and Sample

This study used a cross-sectional design. Hong Kong Chinese parents of children with cancer admitted to Hong Kong Children’s Hospital were invited to participate in this study. To be eligible for this study, parents had to (1) be the primary caregiver of the child (father or mother), (2) be able to speak Cantonese and read Chinese, and (3) have a child aged 0–16 years that was diagnosed with cancer at some time in the previous month and currently undergoing active treatment. We excluded parents with cognitive and learning problems.

To identify potential participants, a promotional poster describing the nature and purpose of this study was posted on the noticeboard of the pediatric oncology unit. Parents of children with cancer could then express their willingness to join the study by calling the mobile phone of the research assistant. A briefing session for interested parents of children with cancer was then conducted at the pediatric oncology unit. 

The sample size was determined by the number of participants that were required for multiple regression analysis using the following formula: *n* > 50 + 8m (where m = the number of independent variables) [17]. The calculated sample size required at least 114 participants, as there were 8 independent variables (parental marital status, cancer type, treatment received, coping strategies used, resilience level, depressive symptom, perceived social support, and anxiety). 

### 2.2. Data Collection Procedures

Eligible parents of children with cancer were recruited between January 2020 and March 2022 from Hong Kong Children’s Hospital. Before the study started, ethical approval was obtained from the Institutional Review Board of the Hong Kong Polytechnic University (HSEARS20190805001). At study enrolment, an information sheet explaining the purpose and nature of this study and associated potential benefits and harms was provided to eligible parents. Written informed consent was obtained from parents whose eligibility was confirmed. After the baseline assessment, all parents were given an information pamphlet about mental health (Chinese version) published by the Central Health Education Unit of the Hong Kong Department of Health. Professional counselling hotline numbers were also provided in the information pamphlet.

#### Data Analysis

SPSS for Windows (SPSS version 26.0; IBM Corp., Armonk, NY, USA) was used for data analysis. Descriptive statistics were used to calculate the mean, standard deviation, and frequency of the demographic and health-risk behavior data. Chi-square tests for independence were used to compare any differences in health-risk behaviors between participants with and without NCDs.

### 2.3. Measures

#### 2.3.1. Connor-Davidson Resilience Scale (CD-RISC)

Parents’ resilience was measured with the CD-RISC, which is a self-rating scale designed by Connor and Davidson [18]. The scale has been widely used in clinical practice and treatment-outcome research [19]. The Chinese version of the CD-RISC was translated by Yu and Zhang [20]. It contains a total of 25 items divided into 3 subcategories: (1) tenacity, (2) strength, and (3) optimism. Each item such as “think of myself as strong person” is rated on a 5-point Likert scale (4 = always, 3 = often, 2 = sometimes, 1 = seldom, 0 = never). The total possible scores for the scale ranges from 0 to 100, with a higher score indicating greater resilience. The scale was found to have good internal consistency (α = 0.84; subcategories α = 0.61–0.88), showing that the scale is reliable.

#### 2.3.2. EuroQoL 5-Dimension 5-Level (EQ-5D-5L)

The EQ-5D-5L was used to measure parents’ quality of life. This is a generic health status measure developed by the EuroQoL Group to assess quality of daily life [21]. The scale offers a descriptive system for self-reported health status and comprises five dimensions: mobility, self-care, usual activities, pain/discomfort, and anxiety/depression. Parents were asked to rate each dimension such as “I have no problems doing my usual activities” on a 5-point scale (1 = unable/extreme problems, 2 = severe problems, 3 = moderate problems, 4 = slight problems, and 5 = no problem). The total possible score for the scale range from 5 to 25, with a higher score indicating better quality of life. The psychometric properties of the Chinese version have been tested and the scale was found to be a valid, reliable, and sensitive measure to assess health-related quality of life [22].

#### 2.3.3. Coping Behavior Checklist (CBC)

The CBC was used to identify the coping strategies used by parents [23]. The CBC was developed based on Lazarus and Folkman’s theory of cognitive appraisal, stress, and coping [24]. The CBC consists of eight items that document how people believe they would cope when faced with a very stressful event, such as “seeking social support”. In this study, participating parents were asked to indicate the frequency of occurrence of the behavior described in each item by selecting one of the following options: “hardly ever,” “sometimes,” or “often,” which are scored from 0 to 2. The total possible score ranges from 0 to 16 and scores of 0–8 and 9–16 correspond to emotion-focused and problem-focused responses, respectively. The psychometric properties of the CBC have been empirically tested, and showed the scale had an α coefficient of 0.71, construct validity of r = −0.58 (*p* < 0.01), and test-retest reliability of r = 0.83 (*p* < 0.01). The results of confirmatory factor analysis further supported the construct validity of the CBC [23]. 

#### 2.3.4. Center for Epidemiologic Studies Depression Scale (CES-D) 

Parents’ depressive symptoms were evaluated with the CES-D, which consists of 20 items [25]. Parents were asked to rate the frequency of each symptom during the past week on a 4-point Likert scale representing “rarely” (less than 1 day), “some” (1–2 days), “occasionally” (3–4 days) and “most” (5–7 days), which are scored from 0 to 3. A sample item is “I felt like I was too tired to do things”. Total possible scores range from 0 to 60, and higher scores indicate a greater risk for depression. The cut-off point is fixed at 16; a score of ≥16 indicates an individual demonstrates some depressive symptoms. The construct validity of the Chinese-language version of the CES-D was tested in a Hong Kong Chinese context and showed excellent construct validity [26]. In this study, the CES-D was found to have good internal consistency (α = 0.93). Appropriate item-total correlations were also found for the CES-D, with values ranging from 0.49 to 0.77.

#### 2.3.5. Chinese Version of the State Anxiety Scale for Adults (C-SAS-A)

Parents’ state anxiety was measured using the C-SAS-A. The scale consists of 20 items scored from 1 to 4, with possible scores ranging from 20 to 80. A sample item is “I feel anxious”. Higher scores indicate greater anxiety. The psychometric properties of the C-SAS-A have been empirically tested [27,28,29]. The internal consistency was found to be good (α = 0.90) [27]. The factor structure of the scale was also confirmed by exploratory and confirmatory factor analyses [27,28]. These findings suggested that the scale can be used as an objective self-report assessment tool to measure anxiety among Chinese adults.

#### 2.3.6. Multidimensional Scale of Perceived Social Support (MSPSS) 

The MSPSS was used to measure parents’ perceived social support. The MSPSS is a 12-item scale with responses on a seven-point scale (from 1 = strongly disagree to 7 = strongly agree) that measures three sources of support: family, friends, and significant others [30]. A sample item is, “My family is willing to help me make decisions”. The psychometric properties of the Chinese version of the MSPSS have been tested [29], and the scale demonstrated excellent internal consistency (α = 0.89). In terms of construct validity, the Chinese version of the MSPSS was negatively correlated with depression (r = −0.16, *p* < 0.01) and anxiety (r = −0.11, *p* < 0.05) as assessed by the General Health Questionnaire. Concurrent validity was demonstrated by the positive association between the MSPSS and the Lubben Social Network Scale (r = 0.41, *p* < 0.01) [31].

### 2.4. Data Analysis

SPSS version 26.0 for Windows (IBM Corp., Armonk, NY, USA) was used to analyze the data. Descriptive statistics were used to calculate the mean, standard deviation (SD), and range of the scores for the different scales. The internal consistencies of the instruments used in this study were determined by calculating their Cronbach’s α. The interrelationships among scores for the CD-RISC, CES-D, CBC, C-SAS-A, MSPSS, and EQ-5D-5L, and participants’ demographic data were investigated using Pearson’s product moment correlation coefficients. A one-way between-groups analysis of variance (ANOVA) was used to assess the mean CD-RISC, CES-D, CBC, C-SAS-A, MSPSS, and EQ-5D-5L scores by parental marital status, child’s cancer type, and coping strategies used. Multiple regression analysis was used to explore whether resilience, depressive symptoms, anxiety, and perceived social support were associated with parents’ quality of life, while controlling for the possible effects of their marital status, coping strategy used, and their child’s cancer type. 

## 3. Results

Between January 2020 and March 2022, 119 parents of children with cancer consented to participate in this study. Participants included 21 fathers and 98 mothers with a mean age of 40.9 years (Table 1). Eleven participants (9.2%) were from single-parent families. Most children had a diagnosis of blood cancer (62.2%) and had received a single treatment (69.7%). The classification of cancer types (blood cancers vs. solid tumors) refers to our previous study [32]. In addition, over half of the parents adopted emotion-focused coping strategies (61.3%). Parents’ mean scores for resilience, depressive symptoms, anxiety, perceived social support, and quality of life are shown in Table 1. 

Table 2 presents the interrelationships among the CD-RISC, CBC, CES-D, C-SAS-A, MSPSS, and EQ-5D-5L scores and demographic characteristics. With reference to Cohen [33], correlation coefficients of 0.10–0.29, 0.30–0.49, and 0.50–1.0 were typically interpreted as small, medium, and large correlations, respectively. There were statistically significant strong positive correlations between the following: CD-RISC and EQ-5D-5L scores (r = 0.60, *p* < 0.01); CES-D and C-SAS-A scores (r = 0.58, *p* < 0.01); CBC and CD-RISC scores (r = 0.53, *p* < 0.01); and CES-D scores and parental marital status (r = 0.53, *p* < 0.01). There were statistically significant strong negative correlations between CD-RISC and CES-D scores (r = −0.57, *p* < 0.01) and between CES-D and EQ-5D-5L scores (r = −0.57, *p* < 0.01). There were statistically significant medium positive correlations between the following: CES-D scores and child’s cancer type (r = 0.44, *p* < 0.01); CES-D scores and treatment received (r = 0.39, *p* < 0.01); C-SAS-A scores and child’s cancer type (r = 0.49, *p* < 0.01); C-SAS-A scores and treatment received (r = 0.45, *p* < 0.01); and MSPSS and CD-RISC scores (r = −0.42, *p* < 0.01). Furthermore, there were statistically significant medium negative correlations between the following: CES-D and CBC scores (r = 0.32, *p* < 0.01); C-SAS-A and CD-RISC scores (r = −0.46, *p* < 0.01); MSPSS scores and parental marital status (r = −0.33, *p* < 0.01); EQ-5D-5L scores and parental marital status (r = −0.45, *p* < 0.01); EQ-5D-5L scores and child’s cancer type (r = −0.32, *p* < 0.01); and EQ-5D-5L and C-SAS-A scores (r = −0.42, *p* < 0.01).

Based on the results of these correlation analyses, we performed between-groups ANOVA to compare the means scores for the CD-RISC, CES-D, C-SAS-A, MSPSS, EQ-5D-5L by participants’ demographic characteristics and coping strategies used. The results (Table 3) showed that participants from single-parent families reported statistically significantly lower levels of resilience (*p* < 0.001), more depressive symptoms (*p* < 0.001), less perceived social support, and poorer quality of life (*p* < 0.001) than participants who lived with their partners (married). The results also showed that parents of children who were diagnosed with a solid tumor or had received multiple cancer treatments reported statistically significantly lower levels of resilience (*p* < 0.001), more depressive symptoms (*p* < 0.001), higher state anxiety scores, and poorer quality of life (cancer type, *p* < 0.001; treatment received, *p* = 0.004) than parents of children who were diagnosed with blood cancer or that had received a single treatment. In addition, parents who adopted problem-focused coping strategies reported statistically significantly higher levels of resilience (*p* < 0.001), fewer depressive symptoms (*p* < 0.001), lower state anxiety scores, greater perceived social support, and better quality of life (*p* < 0.001) than those who adopted emotion-focused coping strategies.

Table 4 shows the summary results of a multiple regression analysis. The overall model explained 47% of the variance. After controlling for the possible effects of parent marital status, child’s cancer type, treatment received, and coping strategies used, we found that the R^2^ change value was 0.12; that is, resilience, depressive symptoms, anxiety, and social support explained an additional 12% of the variance in quality of life. When all variables were entered into the model, only resilience made a statistically significant contribution (*p* = 0.01) with a β coefficient of 0.27, indicating that resilience was associated with quality of life.

## 4. Discussion

This study examined resilience, way of coping, perceived social support, psychological well-being, and quality of life among Hong Kong Chinese parents of children with cancer, an area that is underexplored in the literature. The results of this study indicated that almost half (47.9%) of the parents were at or above the cutoff score of 16 on the CES-D, indicating the presence of some depressive symptoms. These findings were consistent with a previous meta-analysis conducted across 14 countries that showed that many parents of children with cancer were potentially at risk for depression [6]. However, the Chinese parents in our study were more vulnerable to depressive symptoms and had a higher risk for depression than non-Chinese parents in the previous meta-analysis [6]. A possible reason for this discrepancy may be differences in cultural beliefs between Chinese and Western people. Because of the influence of Confucianism and fatalism, many Chinese parents may believe that everything in the human world has already been determined and that nothing can be executed to change the situation [14] including the cancer prognosis of their child. Therefore, they seldom asked for social support and tended to escape from problems rather than searching for solutions when they experienced difficulties, making them more susceptible to psychological distress. Lazarus and Folkman [24] claimed that people may adopt more emotion-focused than problem-focused coping strategies when they perceived a lack of control over a stressful situation. The results of this study showed that many Chinese parents (61.3%) adopted emotion-focused coping strategies and reported statistically significantly lower levels of resilience, more depressive symptoms, higher state anxiety scores, less perceived social support, and poorer quality of life than those who adopted problem-focus coping strategies.

The mean state anxiety score of parents in this study was relatively high. These scores (mean = 44.36, SD = 8.8) were higher than those in a previous study that used the same scale to measure state anxiety among Chinese parents whose children underwent surgery (mean = 37.4, SD = 9.47). These results suggest that the diagnosis of cancer in children causes major worries, upset, and uncertainty among parents.

Our findings revealed that children who were diagnosed with a solid tumor or that had received multiple cancer treatments were associated with more depressive symptoms, higher state anxiety, and poorer quality of life in parents. This is understandable as children who are diagnosed with a solid tumor (e.g., brain tumor, osteosarcoma) may have a comparatively longer treatment time, lower survival rate, and more negative physical and psychological sequelae than those with blood cancer [32,34] In addition, children that had received more than one cancer treatment may experience greater occurrence and severity of treatment-related symptoms [35], which may have an indirect negative impact on their parents’ psychological well-being.

Consistent with previous studies [36,37], this study revealed that single parents of children with cancer reported significantly lower resilience, more depressive symptoms, less perceived social support, and poorer quality of life than parents who lived with their partners. For a single parent, having a child with cancer may result in greater challenges such as financial burden, social isolation, and diminished social support [37]. 

Our results showed that there was a high negative correlation between resilience and depressive symptom scores, as well as a high positive correlation between resilience and quality of life. Moreover, there was a medium negative correlation between resilience and state anxiety, as well as a medium positive correlation between resilience and perceived social support. These findings were consistent with previous studies conducted in Western countries that showed greater parental resilience was associated with better psychosocial outcomes and quality of life [10,11]. The results of our multiple regression analysis showed that resilience was the only factor that was associated with quality of life in parents of children with cancer. The findings of this study further support the literature that showed parents with a higher resilience level had greater strength and an ability to overcome stress and adversity throughout their children’s cancer trajectory, which consequently empowered them and maintained their psychological well-being and quality of life. 

This study provided further support to the existing literature that showed resilience is an important factor for maintaining psychological well-being and quality of life among parents of children with cancer. Assessing resilience in parents of children with cancer is therefore crucial for a thorough understanding of their responses to stress and adversity, which is a prerequisite for designing appropriate psychological interventions to enhance their resilience. Most importantly, such interventions should be implemented by healthcare professionals at an early stage when the parents first learn of their children’s cancer diagnosis to help parents better manage their children’s health condition, navigate adversity throughout their children’s cancer trajectory, and consequently improve their psychological well-being and quality of life.

This study revealed that the adoption of emotion-focused coping strategies among Chinese parents may make them more vulnerable to psychological distress. In addition, we found that single parents and parents of children who were diagnosed with a solid tumor or that received multiple cancer treatment were associated with more depressive symptoms, higher state anxiety, and poorer quality of life. More attention and psychological support should therefore be given to these vulnerable groups.

This study had some limitations. First, this study was the cross-sectional design, which means we were unable to infer causal relationships between parental resilience and quality of life. It is recommended that a further longitudinal study should be conducted to explore whether resilience can be a predictor for quality of life among parents of children with cancer. It is also worth evaluating the process of resilience properties over time, and the relationship between changing resilience resources and their outcomes. Secondly, there is a paucity of qualitative data to thoroughly characterize the concerns or worries of parents of children with cancer that cause them to feel depressed and anxious, jeopardizing their quality of life. It is recommended that qualitative studies should be conducted to explore the challenges experienced by parents and investigate their adjustment and adaptation to their child’s illness. A third limitation of this study is that, given all eligible participants were invited by the investigators to participate in the study, this introduced a potential risk of self-selection bias, which could affect the representativeness and generalizability of results. 

In addition to the limitations and recommendations discuss above, future research is worthy of exploring how Chinese culture (i.e., Confucianism or fatalism) affects the resilience and quality of life of parents of children with cancer. Furthermore, since cancer diagnosis is characterized by uncertainty about the future, it is crucial for future research to examine the role of hope in buffering psychological distress and resilience in parents of children with cancer. Last but not least, future research should examine the caregiver burden, which can have a significant impact on parents’ quality of life.

## 5. Conclusions

The findings of this study make an important contribution to improving healthcare professionals’ understanding about psychological well-being and quality of life among Chinese parents of children with cancer. In addition, this study supports the existing literature that suggests resilience is an important factor associated with parents’ quality of life. Appropriate resilience-enhancing interventions should be developed, implemented, and evaluated for parents of children with cancer, in particular for those with a low resilience level at the time of their child’s cancer diagnosis.

## Figures and Tables

**Table 1 ijerph-20-05765-t001:** Demographic characteristics of the parents and children (*n* = 119).

	Frequency	%
Sex (parents)		
Male	21	17.6
Female	98	82.4
Sex (children)		
Male	56	47.1
Female	63	52.9
Parental marital status		
Married	108	90.8
Single-parent family	11	9.2
Parents’ educational attainment		
Primary school or below	3	2.5
Lower secondary school	27	22.7
Upper secondary school	55	46.2
Tertiary education	34	28.6
Religion		
Yes	81	68.1
No	38	31.9
Cancer type		
Blood cancer	74	62.2
Solid tumor	45	37.8
Treatment received		
Single treatment	83	69.7
Multiple treatments	36	30.3
Coping strategy		
Emotion-focused	73	61.3
Problem-focused	46	38.7
	**Mean**	**Standard Deviation**
Ages (parents)	40.9	6.5
Ages (children)	8.4	4.6
Resilience	63.4	9.5
Depressive symptoms	17.1	6.2
Anxiety	44.4	8.8
Social support	65.7	9.1
Quality of life	19.6	2.9

**Table 2 ijerph-20-05765-t002:** Intercorrelation coefficients among the scores of the CD-RISC, CBC, CES-D, C-SAS-A, MSPSS, EQ-5D-5L, and demographic characteristics (*n* = 119).

	A	B	C	D	E	F-1	F-2	F-3	F-4	G	H	I	J	K	L	M	N	O
Age, parents (A)	1																	
Age, children (B)	0.44 *	1																
Sex, parents (C)	−0.03	−0.04	1															
Sex, children (D)	−0.30 *	−0.26 *	−0.13	1														
Parental marital status (E)	0.14	0.15	−0.07	0.01	1													
Parents’ educational attainment																		
Primary school or below (F-1)	0.11	0.14	0.06	0.04	0.13	1												
Lower secondary school (F-2)	0.03	−0.01	−0.04	0.14	0.03	−0.08	1											
Upper secondary school (F-3)	0.07	0.06	0.10	−0.17	−0.12	−0.14	−0.20	1										
Tertiary education (F-4)	−0.14	−0.11	−0.09	0.03	0.05	−0.10	−0.14	−0.19	1									
Religion (G)	0.01	0.03	0.10	−0.07	0.09	−0.11	−0.19 *	0.19 *	0.01	1								
Cancer type (H)	0.21 *	0.22 *	−0.08	−0.13	0.20	−0.01	0.07	0.04	−0.11	−0.08	1							
Treatment received (I)	0.18 *	0.20 *	−0.11	−0.18 *	0.04	0.01	−0.09	0.12	−0.05	0.09	0.58 *	1						
CD-RISC (J)	−0.21 *	−0.20 *	0.10	0.01	−0.45 *	−0.06	−0.20	0.13	0.15	0.17	−0.32 *	−0.06	1					
CBC (K)	−0.11	−0.04	−0.05	−0.04	−0.19 *	0.09	−0.22 *	0.12	0.03	0.01	−0.01	−0.03	0.53 *	1				
CES-D (L)	0.13	0.18 *	−0.10	−0.03	0.53 *	−0.04	0.12	−0.08	−0.00	−0.13	0.44 *	0.39 *	−0.57 *	−0.32 *	1			
C-SAS-A (M)	0.03	0.16	0.01	0.01	0.14	−0.08	0.16	−0.03	−0.08	−0.19 *	0.49 *	0.45 *	−0.46 *	−0.31 *	0.58 *	1		
MSPSS (N)	−0.19 *	−0.19 *	−0.05	−0.01	−0.33 *	−0.11	−0.16	0.10	0.08	0.06	−0.18 *	−0.02	0.42 *	0.27 *	−0.31 *	−0.16	1	
EQ-5D-5L (O)	−0.07	−0.07	0.01	−0.04	−0.45 *	−0.01	−0.19 *	0.21 *	−0.11	0.18 *	−0.32 *	−0.26 *	0.60 *	0.42 *	−0.57 *	−0.42 *	0.32 *	1

Abbreviations: CD-RISC, Connor-Davidson resilience scale; CBC, Coping Behaviour Checklist; CES-D, Center for Epidemiologic Studies Depression Scale; C-SAS-A, Chinese version of the State Anxiety Scale for Adults; MSPSS, Multidimensional Scale of Perceived Social Support; EQ-5D-5L, EuroQoL 5-Dimension 5-level; * Significant at *p* < 0.01.

**Table 3 ijerph-20-05765-t003:** Between-groups ANOVA to compare the means scores of CD-RISC, CBC, CES-D, C-SAS-A, MSPSS, and EQ-5D-5L of participants with different parental marital status, cancer type, treatment received, and coping strategy (*n* = 119).

		CD-RISC		CES-D		C-SAS-A		MSPSS		EQ-5D-5L	
	(*n*)	Mean (SD)	*p*	Mean (SD)	*p*	Mean (SD)	*p*	Mean (SD)	*p*	Mean (SD)	*p*
Parental marital status			<0.001		<0.001		0.127		<0.001		<0.001
Married	108	65.4 (9.58)		15.38 (4.42)		43.79 (8.79)		67.21 (9.28)		20.03 (2.69)	
Single-parent family	11	43.09 (2.78)		33.82 (6.94)		50.00 (7.11)		50.55 (9.64)		15.45 (1.29)	
Cancer Type			<0.001		<0.001		<0.001		0.054		<0.001
Blood cancer	74	66.99 (9.72)		13.59 (4.85)		39.43 (7.51)		67.84 (9.16)		20.32 (2.75)	
Solid tumor	45	57.42 (9.90)		22.82 (7.09)		52.47 (8.31)		62.11 (9.14)		18.42 (2.80)	
Treatment received			<0.001		<0.001		<0.001		0.108		0.004
Single treatment	83	66.2 (9.02)		14.4 (5.13)		40.57 (8.09)		67.14 (9.92)		20.11 (2.85)	
Multiple treatment	36	56.83 (8.59)		23.17 (5.85)		53.11 (8.46)		62.28 (9.14)		18.44 (2.74)	
Coping strategy			<0.001		<0.001		0.001		<0.001		<0.001
Emotion-focused	73	57.36 (8.89)		19.67 (7.82)		47.48 (9.10)	0.001	62.44 (9.59)		18.64 (2.68)	
Problem-focused	46	72.91 (9.16)		12.98 (3.98)		39.41 (7.54)		70.80 (9.87)		21.13 (2.61)	

Abbreviations: CD-RISC, Connor-Davidson resilience scale; CES-D, Center for Epidemiologic Studies Depression Scale; C-SAS-A, Chinese version of the State Anxiety Scale for Adults; MSPSS, Multidimensional Scale of Perceived Social Support; EQ-5D-5L, EuroQoL 5-Dimension 5-level; SD, Standard deviation.

**Table 4 ijerph-20-05765-t004:** Summary of multiple regression for variables predicting quality of life (*n* = 138).

Variable Predicting Quality of Life	*B*	*SE B*	*β*	*p-*Value
Step 1				
Parent marital status	−3.18	0.82	−0.32	<0.001
Cancer type	−1.21	0.48	−0.20	0.06
Treatment received	−0.20	0.59	−0.03	0.73
Coping strategy	2.10	0.46	0.35	<0.001
Step 2				
Parent marital status	−1.7	0.90	−0.17	0.07
Cancer type	−0.18	0.52	−0.03	0.64
Treatment received	0.21	0.56	0.03	0.71
Coping strategy	0.84	0.50	0.14	0.10
Resilience	0.06	0.02	0.27	0.01
Depressive symptoms	−0.06	0.03	−0.20	0.06
Anxiety	−0.02	0.02	−0.09	0.34
Social support	0.01	0.02	0.03	0.71
R^2^ = 0.47				
Adjust R^2^ = 0.44				
R^2^ change = 0.12				

*B* = unstandardized coefficient; *SE B* = standard error of unstandardized coefficient; *β* = standardized coefficient.

## Data Availability

The data presented in this study are available on request from the corresponding author. The data are not publicly available due to privacy and ethical concerns.

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
