# Peer review of "The Association of Resilience with Way of Coping, Psychological Well-Being and Quality of Life in Parents of Children with Cancer"

_ijerph, 2023, doi:10.3390/ijerph20105765_

Round 1
Reviewer 1 Report
Chung and his group report a cross-sectional study which explored the association of resilience with way of coping, quality of life and psychological well-being of Chinese parents of children with cancer. Although a relevant study for the population of Hong Kong, it is not a novel study. However, it definitely serves as a validation of previous literature.
Their findings support the existing literature showing that resilience is an important factor for maintaining psychological well-being and quality of life among parents of children with cancer. Authors have reported the presence of depressive symptoms and anxiety, and poor life quality in substantial percent of the parents. It is even higher in parents of children diagnosed with a solid tumor as well as single parents. Finally, they suggest that adoption of problem-focused coping strategies should be given preference by clinician compared to the emotion-focused.
Authors have provided adequate basis for the study, conducted well-designed experiments and applied appropriate analytical approaches, and cited proper references through the manuscript.
Overall, the current manuscript is presented in a well-structured manner and has potential to contribute to the clinical field. There are certainly few limitations which authors have identified and properly addressed in the discussion.
I do not have any concern or suggestion. However, I do have a question: do authors have any plans or any ongoing study to conduct longitudinal investigations. If not, can authors briefly describe some of the key requirements or characteristics of the study design, e.g., future study should assess the course of resilience properties over time, and the relationships between changing resilience resources and its outcome. This could prove valuable for researchers in the field. A major gap is the limited interpretation of the directionality from this and similar studies which ultimately limits the application of these findings to design appropriate strategies or interventions.
Reviewer 2 Report
Thank you for the opportunity to revise this manuscript. The paper deals with an interesting topic, understudied in the regard of the Chinese population.
The paper is clear and well written. Methodology is well explained and adequate.
I provide below some suggestion, with the aim of improving the quality of the paper and its impact to the readers.
Introduction: line 66-67: the authors refer to a review of the literature. The precise reference citation is needed
In the end of the Introduction section the authors could specify their research hypothesis and expected results, according to previous literature. They could also be clearer about their aim. What is the specific dependent variable? Quality of life? If yes, they should provide a theoretical background based on which they decide to use this as dependent variable, instead of, for instance, depression.
Methods: an example item for each questionnaire should be added in the Measures section
The authors should better specify on which basis they decided to categorize cancer type (blood vs solid) and treatment. Did they took methodology from previous studies? If yes, they should provide reference
Limitation and future perspectives.
As regard Limitations, the authors should add the risk of self-selection bias, that can have affected representativeness and generalizability of results. Indeed, participants explicitly called the investigators to ask to participate.
Moreover, I think the authors should add a paragraph regarding the Future perspectives, within the Discussion section of the paper: in this regard, in the paper the authors provide an interesting consideration concerning the role of type of religion and its association with coping styles. Unfortunately, the authors did not investigate this variable among participants. Future studies could better explore this aspect, specifically investigating this variable (for instance Confucianism or fatalism). Even better if in the discussion section they provide some reference in regard of this aspect.
In addition, I think two other variables could be worth investigating in future studies, with the aim of enriching comprehension of this phenomenon. One is Hope. Under many aspects a diagnosis of Cancer is characterized by uncertainty about future. Other studies underline the role of hope in buffering psychological distress and resilience (see for instance https://doi.org/10.1186/s12955-016-0481-z , DOI: 10.1097/NCC.0000000000000753 , https://doi.org/10.3390/ejihpe13010005 )
Finally, another significant variable worth noting is careviger burden of parents. Indeed, caregiver burden can significantly impact people's quality of life) and psychological distress, and this is a relevant topic for parents of children who receive a diagnosis of cancer.
Round 2
Reviewer 2 Report
The revised version of the paper is suitable for publication.
Congratulations to the authors